# Sustainable Green Energy Management: Optimizing Scheduling of Multi-Energy Systems Considered Energy Cost and Emission Using Attractive Repulsive Shuffled Frog-Leaping

Kumaran Kadirgama [1], Omar I. Awad [2,3,4,*] , M. N. Mohammed [2] , Hai Tao [5] and Ali A. H. Karah Bash [6]

[1] Mechanical and Automotive Engineering Technology, Universiti Malaysia Pahang, Pekan 26600, Pahang, Malaysia; kumaran@ump.edu.my
[2] Mechanical Engineering Department, College of Engineering, Gulf University, Sanad 26489, Bahrain; dr.mohammed.alshekhly@gulfuniversity.edu.bh
[3] Mechanical and Electrical Engineering College, Hainan University, Haikou 570228, China
[4] Engineering College, University of Kirkuk, Kirkuk 36001, Iraq
[5] School of Artificial Intelligence, Nanchang Institute of Science and Technology, Nanchang 330099, China; haitao@bjwlxy.edu.cn
[6] Department of Electrical and Electronics Engineering, University of Gaziantep, Gaziantep 27310, Turkey; alikarabash2016@yahoo.com
* Correspondence: omaribr78@gmail.com

**Abstract:** As energy systems become increasingly complex, there is a growing need for sustainable and efficient energy management strategies that reduce greenhouse gas emissions. In this paper, multi-energy systems (MES) have emerged as a promising solution that integrates various energy sources and enables energy sharing between different sectors. The proposed model is based on using an Attractive Repulsive Shuffled Frog-Leaping (ARSFL) algorithm that optimizes the scheduling of energy resources, taking into account constraints such as capacity limitations and environmental regulations. The model considers different energy sources, including renewable energy and a power-to-gas (P2G) network with power grid, and incorporates a demand–response mechanism that allows consumers to adjust their energy consumption patterns in response to price signals and other incentives. The ARSFL algorithm demonstrates superior performance in managing and minimizing energy purchase uncertainty compared to the particle swarm optimization (PSO) and genetic algorithm (GA). It also exhibits significantly reduced execution time, saving approximately 1.59% compared to PSO and 2.7% compared to GA.

**Keywords:** renewable energy; electricity-to-gas technology; optimal dispatch; multi-energy system; energy hub

## 1. Introduction

The increasing demand for energy and the need for sustainability have led to the development of multi-energy systems (MES) that integrate various energy resources to achieve efficient and environmentally friendly energy management [1,2]. Sustainable power-to-gas (P2G) technology is a promising solution to enable the large-scale integration of renewable energy sources into the grid. P2G technology allows excess electricity generated from renewable sources, such as wind and solar, to be converted into hydrogen or methane gas through electrolysis [3].

In recent years, the research on electricity-to-gas technology (P2G) has become more and more mature. The combination of P2G and energy interconnection has enabled the conversion from electric energy to natural gas, which can effectively enhance the coupling of multi-energy coupling systems. In addition, P2G technology can significantly improve system performance with the ability to absorb renewable energy. There are mathematical models for optimizing the operation of a multi-energy system consisting of electricity,

gas, and heat networks. A model incorporating demand-side management and energy storage to reduce emissions and energy costs, while ensuring energy supply reliability, has been proposed by Li et al. [4]. A stochastic programming model for optimizing the design and operation of a multi-energy system has been presented by Han et al. [5]. The model considers uncertainties in energy demand, renewable energy supply, and energy prices, and aims to minimize the total system cost while ensuring energy supply reliability. Ref. [6] proposes a framework for multi-criteria decision-making in sustainable energy management. The framework incorporates economic, environmental, and social criteria, and can be used to evaluate the sustainability of different energy management strategies. Ref. [7] provides a review of mathematical models and solution techniques for optimizing the integration of renewable energy sources in multi-energy systems. The review highlights the importance of considering the interaction between different energy sources and technologies and the need for robust optimization methods to account for uncertainties. Ref. [8] provides a comprehensive review of multi-energy system planning and optimization models, methods, and applications. The review covers a range of topics, including energy hub models, power flow models, co-simulation models, and optimization models, and highlights the importance of considering economic, environmental, and social objectives in energy system planning. Ref. [9] provides an overview of the concept of multi-energy systems and their potential for improving energy efficiency and reducing greenhouse gas emissions. Further details are given on the current state of research on multi-energy systems, including modeling techniques, optimization methods, and case studies.

Many researchers have proposed various optimization techniques for MES scheduling such as Mixed Integer Linear Programming (MILP), Particle Swarm Optimization (PSO), Fuzzy logic algorithm, Frog-Leaping algorithm, and Genetic Algorithms (GA) [10–14]. However, these methods have limitations, such as high computational complexity, inability to handle uncertainties, and lack of collaboration among stakeholders. Ref. [15] proposes a mixed-integer linear programming model for optimizing the design and operation of a multi-energy systems model through a case study of smart stadiums in China. Ref. [16] presents a multi-objective optimization model for sustainable energy management in a microgrid system considering both economic and environmental objectives. Ref. [17] proposes a mixed-integer linear programming model for optimizing multi-energy systems that considers both thermal and electric energy storage. Ref. [18] focuses on the advantages of priority regulation of pumped storage for the carbon emission-oriented co-scheduling of hybrid energy systems. The scheduling method takes into account carbon emissions and the operation of a pumped storage system to optimize the operation of a hybrid energy system. Ref. [19] propose a multi-objective optimization model for scheduling the operation of an offshore micro-integrated energy system that considers natural gas emissions. The model considers multiple objectives, including economic efficiency and environmental impact. Ref. [20] presents a stochastic optimal scheduling model for multi-microgrid systems that considers emissions. The constrained model takes into account the stochastic nature of renewable energy sources and the uncertain demand for electricity. Ref. [21] optimizes the scheduling of residential battery energy storage systems to reduce both cost and emissions, and takes into account the fluctuation of electricity demand and solar power output. Ref. [22] proposes a multi-objective generation scheduling model for an integrated energy system that considers economic and environmental factors. The model uses a fuzzy-based surrogate with a trade-off approach to balance conflicting objectives. Ref. [23] presents a stochastic multi-objective optimization model for scheduling the operation of microgrids that includes battery energy storage systems. The model considers economic and environmental objectives and uses a stochastic programming method to deal with the uncertainties in renewable energy sources. Ref. [24] propose a tri-objective optimization model for scheduling the operation of a smart energy hub system that includes schedulable loads. The model considers multiple objectives, including economic efficiency, environmental impact, and social welfare. Ref. [25] gives a multi-objective complementary scheduling model for a hydro-thermal-RE power system using a multi-objective hybrid

grey wolf algorithm to optimize the operation of the system while considering multiple objectives, including economic efficiency and environmental impact. However, managing MES is complex due to the diverse nature of the energy sources and the different energy demands of various sectors. There is a need for comprehensive and integrated optimization approaches in managing Multi-Energy Systems (MES) that consider the diverse nature of energy sources, the uncertain and intermittent characteristics of renewable energy, and computational complexity.

This paper proposes a collaborative scheduling model using Attractive Repulsive Shuffled Frog-Leaping (ARSFL) for MES that aims to promote sustainable and efficient energy management while reducing emissions. The model focuses on integrating various energy sources, such as renewable energy, power grids, and storage systems, while considering constraints and objectives related to energy cost, emission reduction, and sustainability. Via the establishment of constraint conditions and a case analysis, the series characteristics of the energy hub are also fully utilized. The proposed model is validated according to hourly forecasted electricity, cooling, heat load, and new energy output data, combined with the coupling relationship of the energy hub, to reduce operating costs as much as possible, and make the energy distribution more reasonable. The research presented in this paper provides a valuable contribution to the field of sustainable energy management by proposing a collaborative scheduling model for MES with reduced emissions. The proposed model can serve as a valuable tool for decision-makers in the energy sector seeking to promote sustainability and efficiency in their operations.

## 2. Related Work

### 2.1. Sustainable P2G Technology Energy Management

Sustainable energy management involves managing energy resources and systems in a way that reduces environmental impacts, enhances energy security and reliability, and promotes economic growth. It includes the adoption of renewable energy sources and energy efficiency measures, and reductions in greenhouse gas emissions. The theoretical model for sustainable P2G technology energy management is a circular economy model, as shown in Figure 1. Circular economy is a regenerative system that aims to minimize waste and maximize the use of resources by keeping materials in use and reducing the consumption of finite resources. The circular economy model can be applied to P2G technology by using renewable energy sources to power the electrolysis process, producing hydrogen or SNG for energy storage and transportation, and then using the stored energy to power renewable energy systems or replace fossil fuels in industrial processes.

The circular economy model also involves the reuse and recycling of materials and the recycling of hydrogen or SNG after use. Ref. [26] discuss the risk-based performance of power-to-gas (P2G) storage technology in an energy hub system, specifically considering downside risk constraints. The study evaluates the feasibility and economic benefits of integrating P2G with the energy hub system using a risk-constrained optimization model. Ref. [27] analyzes the application of P2G technology in the road transport system of South Africa. The authors evaluate the potential of P2G technology to reduce greenhouse gas emissions and dependence on imported oil. Ref. [28] explores the decarbonization potential of the European electricity system with synthetic methane produced through P2G technology. This features a techno-economic analysis of the feasibility of using P2G technology to integrate renewable energy sources into the electricity system. Ref. [29] conducts a techno-economic analysis of a gas-to-power distributed generation planning system for grid stability and environmental sustainability in Nigeria. It evaluates the economic viability and environmental impact of using gas-to-power technologies, such as P2G, for distributed generation in Nigeria. Ref. [30] proposes an energy management system for solar–hydrogen microgrids that incorporates vehicle-to-grid and P2G transactions. It gives a framework for managing energy flows in a microgrid that includes solar power generation, hydrogen production through electrolysis, and vehicle-to-grid and P2G transactions. Ref. [31] discusses technology-enabled circular business models for hybrid wind farms,

including integrated wind and solar energy, P2G, and power-to-liquid systems. In addition, it explores the potential for circular business models to improve the economic and environmental sustainability of hybrid wind farms. Ref. [32] presents a decision-making methodology for managing surplus photovoltaic electricity through P2G in combined heat and power (CHP) systems in urban buildings. Ref. [33] evaluates the energy supply system in a multi-energy complementary park using an improved universal generating function method. A model that optimizes the energy supply system in a multi-energy complementary park that includes P2G technology for energy storage and management is given in Ref. [34].

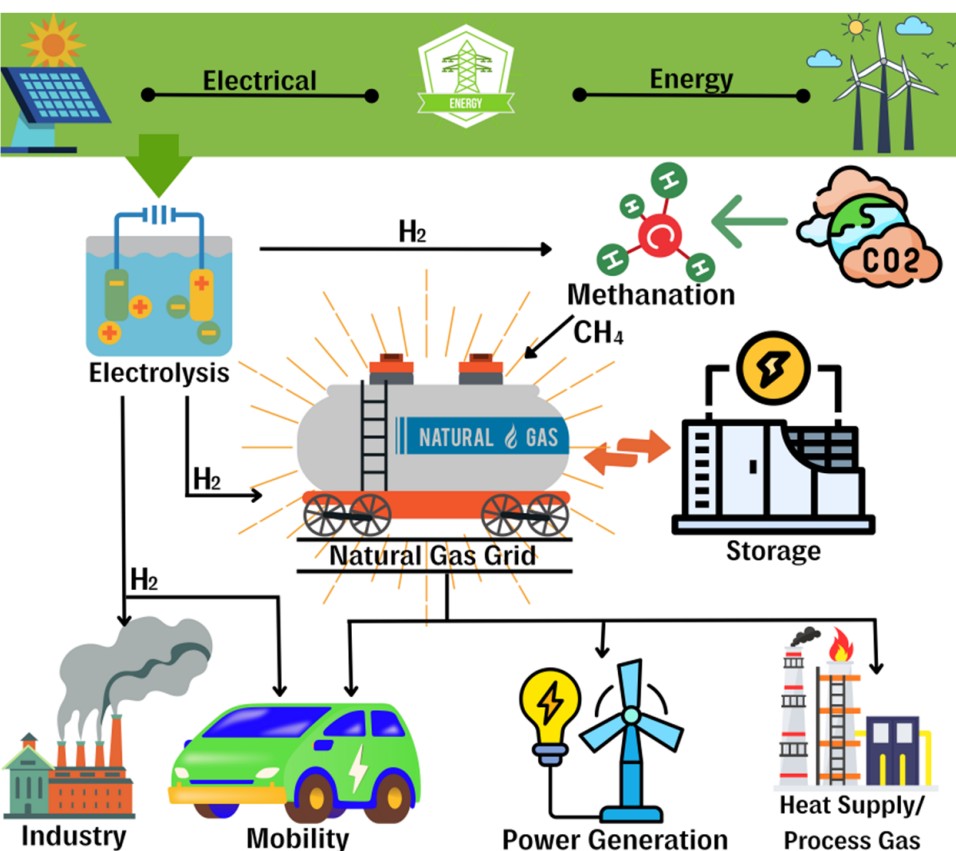

**Figure 1.** Application of sustainable P2G technology energy.

### 2.2. Multi-Energy System Model

Multi-energy systems are energy systems that integrate multiple energy sources and carriers to meet energy demand. They involve the use of various energy sources, such as electricity, natural gas, heating and cooling, and transportation fuels, and allow for flexibility in the use of energy resources. The multi-energy system has an input of multiple energy forms, and features the coupling of different energy equipment types with multiple energy requirements. This paper establishes a universal multi-energy system based on the concept of the energy hub, which is used to describe the exchange and coupling relationship between energy, load, and network in the system [28], as shown in Figure 2, where electric energy and natural gas are the two main energy inputs. The coupling equipment types include P2G equipment, micro-gas turbines, boilers, refrigeration equipment, energy storage equipment, etc. The energy demand is categorized into three categories: electricity, cold and heat.

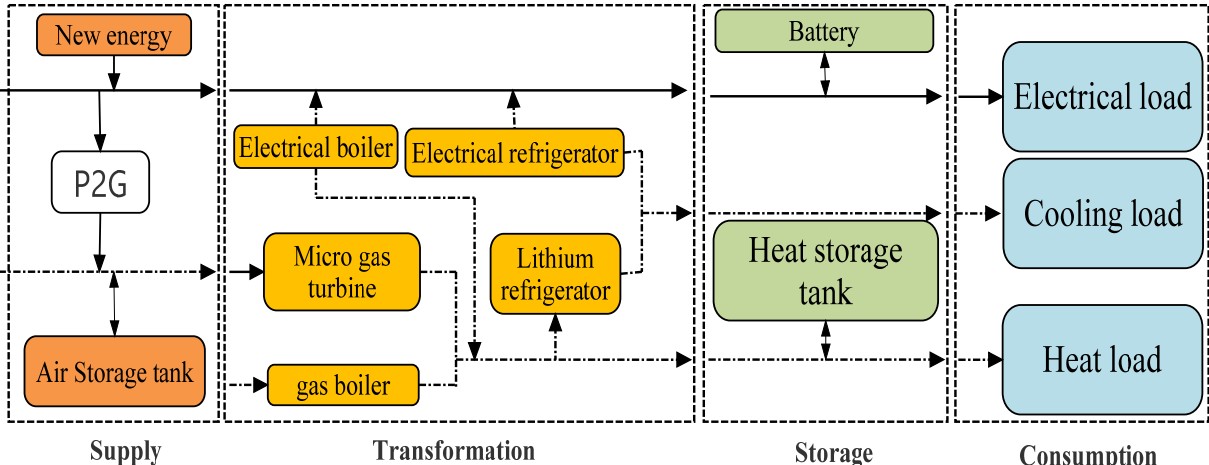

**Figure 2.** Multi-energy system.

In order to reflect the serial characteristics of energy supply, conversion, storage, and consumption, this energy system sets out the four modules of supply, conversion, storage and consumption, as follows.

In the supply module of the gas tank model, the energy input comprises the power grid, the natural gas network, the new energy sources (solar energy and wind energy), and the P2G equipment and gas storage equipment, which can be described as follows:

$$P^t = P^{net} + P^{in} + P^{P2G} \tag{1}$$

$$\begin{bmatrix} P_e^t \\ P_g^t \end{bmatrix} = \begin{bmatrix} P_e^{new} \\ P_g^{new} \end{bmatrix} + \begin{bmatrix} P_e^{new} \\ P_g^s \end{bmatrix} + \begin{bmatrix} -P_e^{P2G} \\ P_g^{P2G} \end{bmatrix} \tag{2}$$

where $P_e^t$, $P_g^t$ is the output of electric energy and natural gas in the module; $P_e^{new}$, $P_g^{new}$ is the input of electric energy and natural gas in the module; $P_e^{new}$, $P_g^s$ is the input of new energy and the gas storage tank.

The energy relationship between the equipment before and after charging and deflation in the gas tank model, assuming constant storage and deflation power during the time period t [21], can be expressed as follows:

$$W_s(t) = W_s(t-1) + P_g^t \tag{3}$$

$$P_g^t = \left[ P_s^{ch} \eta_g^{ch} \mu - \frac{P_s^t}{\eta_g^{dis}} (1 - \mu) \right] \times t \tag{4}$$

where $W_g(t-1)$, $W_g(t)$ is the energy stored in the equipment before and after gas storage or deflation; $P_g^{ch}$, $P_g^{dis}$ is the energy stored or released by the gas storage tank; $\eta_g^{ch}$, $\eta_g^{dis}$ is the gas storage and the efficiency of deflation; $\mu$ variable can take values 1 or 0, which refer to the inflated state and deflated state, respectively.

The conversion module includes micro-gas turbines, electric boilers, gas boilers, electric refrigerators and lithium bromide refrigerators. The relationship between these devices is as follows:

$$P^T = C^T P^t \tag{5}$$

$$
\begin{bmatrix} P_e^T \\ P_{ch}^T \\ P_h^T \end{bmatrix} = \begin{bmatrix} \beta_1 & \gamma\eta_{MT}^e \\ \beta_2\eta_{AC}^{ch} & \delta\eta_{AR}^e\left[\gamma\eta_{MT}^e + (1-\gamma)\eta_{GB}^h\right] \\ \beta_3\eta_{EB}^h & (1-\delta)\left[\gamma\eta_{MT}^e + (1-\gamma)\eta_{GB}^h\right] \end{bmatrix} \begin{bmatrix} P_e^t \\ P_g^t \end{bmatrix} \tag{6}
$$

where C is the coupling matrix; $P_e^T$, $P_c^T$, $P_h^T$ converts the electrical output, cold output and heat output of the module. $\beta_1$, $\beta_2$, $\beta_3$ is the electric energy input, $P_e^t$ is the distribution coefficient of the electric load, electric refrigerator and electric boiler, and the sum of the three is equal to 1; $\gamma$ is the micro-gas turbine, which consumes natural gas input; $P_g^t$ is the proportion coefficient; $\delta$ is the proportion coefficient of the total heat consumed by the refrigerator; $\eta_{AC}^c$ is the refrigeration coefficient of the electric refrigerator; $\eta_{AR}^c$ is the refrigeration coefficient of the lithium bromide refrigerator; $\eta_h^{EB}$ is the heating coefficient of the electric boiler; $\eta_h^{GB}$ is the heating coefficient of the steam boiler; $\eta_e^{MT}$, $\eta_h^{MT}$ refer to the micro-gas turbine's electrical efficiency and heating coefficient, respectively.

The storage module includes electrical storage equipment and heat storage equipment. Gas storage equipment is considered in the supply module. Its energy relationship is as follows:

$$
L = P^T + S \tag{7}
$$

$$
\begin{bmatrix} L_e \\ L_c \\ L_h \end{bmatrix} = \begin{bmatrix} P_e^T \\ P_{ch}^T \\ P_h^T \end{bmatrix} + \begin{bmatrix} P_e^s \\ 0 \\ P_h^s \end{bmatrix} \tag{8}
$$

The storage equipment adopts the battery model, which mainly considers its charging and discharging power and current electricity, without considering its internal charging and discharging circuit process.

The state of charge of the battery is:

$$
SOC(t) = (1-\vartheta)SOC(t-1) + \frac{P_e^t}{E_{ch}} \tag{9}
$$

$$
P_e^t = \left[ P_e^{ch}\eta_e^{ch}\mu - \frac{P_s^t}{\eta_e^{dis}}(1-\mu) \right] \times t \tag{10}
$$

where $SOC(t)$ is the state of charge of the battery at t; $\vartheta$ is the discharge rate of the battery itself; $E_c$ is the battery's rated capacity; $P_e^c$, $P_e^{dis}$ is the battery's stored or released energy; $\eta_e^c$, $\eta_e^{dis}$ is the efficiency of discharge; the $\mu$ variable takes values 1 or 0, which denote the charging state and discharging state, respectively.

The energy relationship of the equipment before and after the charging and release of the heat storage tank can be expressed as follows:

$$
W_h(t) = W_h(t-1) + P_h^t \tag{11}
$$

$$
P_h^t = \left[ P_h^{ch}\eta_h^{ch}\mu - \frac{P_s^t}{\eta_h^{dis}}(1-\mu) \right] \times t \tag{12}
$$

where $W_h(t-1)$, $W_h(t)$ is the stored energy of the equipment before and after heat storage or release; $P_h^c$, $P_h^{dis}$ is the energy stored in or released from the heat storage tank; $\eta_h^c$, $\eta_h^{dis}$ is the heat storage and the efficiency of heat release.

Based on the concept and sequential characteristics of the supply module, conversion module, storage module, and consumption module (electric load, cooling load, and heat load), the relationship can be described as follows:

$$L = C^T \left( P^{net} + P^{in} + P^{P2G} \right) + S \tag{13}$$

For the establishment of models in different cases, we need only optimize the different models according to the above-mentioned modules, and modify the corresponding elements in the energy relationship expression matrix.

## 3. Proposed Method

### 3.1. Objective Function

The minimizing multi-object model of the multi-energy system is as follows:

$$f_{min} = f_{cost} + f_{emission} \tag{14}$$

where $f_{cost}$ represents the energy cost of the system and $f_{emission}$ is gas emissions.

Scheduling multi-energy systems (MES) that include power-to-gas (P2G) to reduce energy costs:

Objective function $f_{cost}$ is

$$\text{minimize} \sum (c_i * E_i + \eta * c_g * G_i) \tag{15}$$

where $E_i$ is the energy consumption of the i-th energy resource; $c_i$ is the energy price of the i-th energy resource; $G_i$ is the amount of hydrogen produced by the P2G system from the excess renewable energy of the i-th energy resource; $c_g$ is the cost of hydrogen production; $\eta$ is the efficiency of the P2G system.

Incorporating power-to-gas (P2G) technology into multi-energy systems (MES) to reduce carbon emissions can be formulated as follows.

Objective function $f_{emission}$ is

$$\text{minimize} \ \Sigma i \ \Sigma t \ (c_i * p_{it} + d_i * p_{ht} + e_i * p_{gta} + f_i * p_{gtr} + g_i * p_e + h_i * p_d) \tag{16}$$

where $c_i$ is the cost of purchasing electricity from the grid at time t; $p_{it}$ is the power purchased from the grid at time t; $d_i$ is the cost of natural gas at time t; $p_{ht}$ is the natural gas consumed for heating at time t; $e_i$ is the cost of hydrogen produced from P2G at time t; $p_{gta}$ is the hydrogen consumed for power generation at time t; $f_i$ is the cost of hydrogen stored in tanks at time t; $p_{gtr}$ is the hydrogen consumed for transportation at time t; $g_i$ is the cost of electricity exported to the grid at time t; $p_e$ is the power exported to the grid at time t; $h_i$ is the penalty cost for not meeting the energy demand at time t; $p_d$ is the power demand at time t.

Constraints

Energy balance:

$$\sum (E_i) = D(t) \tag{17}$$

P2G constraints:

$$G_i \leq M_i * P_i \tag{18}$$

Energy storage:

$$S(t) = S(t-1) + \sum (E_i) - D(t) \tag{19}$$

Energy conversion:

$$E_i \leq \eta_i * P_i + (1 - \delta_i) * S(t-1) - S(t) \tag{20}$$

Demand–response:

$$E_i \leq E_{imax} * DR_i(t) \tag{21}$$

Non-negative constraints:

$$\begin{cases} E_i \geq 0 \\ G_i \geq 0 \\ S(t) \geq 0 \end{cases} \qquad (22)$$

$$\lambda P2G \text{ storage capacity}: \text{pgts} <= c * \text{pgtsmax} \qquad (23)$$

where D(t) is the energy demand at time t; Mi is the maximum hydrogen production capacity of the P2G system for the i-th energy resource; Pi is the excess renewable energy of the i-th energy resource; S(t) is the energy storage level at time t; ηi is the efficiency of the i-th energy conversion process; δi is the energy loss coefficient of the i-th energy conversion process; Eimax is the maximum energy consumption of the i-th energy resource; DRi(t) is the demand–response factor of the i-th energy resource at time t; pgts is the hydrogen stored in tanks at time t; c is the P2G storage capacity coefficient.

### 3.2. Attractive Repulsive Shuffled Frog-Leaping

The Shuffled Frog-Leaping (SFL) algorithm is a metaheuristic optimization algorithm inspired by the behavior of frogs in maintaining their population, with each representing a potential solution. It mimics the natural process of frogs leaping and exchanging information to find optimal solutions to complex problems. The Shuffled Frog-Leaping (SFL) algorithm combines exploration and exploitation to optimize solutions by dividing frogs into subgroups to explore the search space and avoid suboptimal solutions. During the leaping phase, frogs exchange information and perform local searches to exploit promising regions and converge towards better solutions.

The AR-SFLA algorithm enhances the exploration and exploitation abilities of the original SFLA by adding attractive and repulsive forces to guide the frogs towards promising regions of the search space, and prevent premature convergence to suboptimal solutions. The AR-SFL algorithm utilizes an attractive–repulsive mechanism to guide the frogs towards better solutions and maintain population diversity. The attractive force attracts frogs to promising regions, promoting exploration and exploitation. Meanwhile, the repulsive force pushes frogs away from crowded or suboptimal regions, encouraging global exploration. By balancing these forces, the algorithm achieves an effective trade-off between exploration and exploitation, optimizing convergence speed and solution quality. This mechanism ensures efficient navigation of the search space, avoids premature convergence, and fosters population diversity.

The AR-SFLA algorithm can be described as follows:

Step one—Generate an initial population of N frogs with random positions and compute their fitness values;

Step two—Compute the attractive and repulsive forces for each frog based on its position and the best solutions found so far;

Step three—Shuffle and divide the frogs into m subpopulations of k frogs each, and shuffle the frogs within each subpopulation randomly;

Step four—For each subpopulation, apply a local search technique, such as gradient descent, to improve the positions of the frogs;

Step five—Select the k best frogs from all the subpopulations and update their positions based on the best solutions. The updated equation for frogs' positions is:

$$x_i(t+1) x_i(t) + \alpha * f_{attract\,i}(t) + \beta * f_{repuls\,i}(t) + \gamma * \text{rand} * (gbest - x_i(t)) \qquad (24)$$

where α, β, and γ are scaling factors, rand() is a random number between 0 and 1, gbest is the best position, and fattracti(t) and frepulsi(t) are the attractive and repulsive forces acting on frog i at time t, respectively.

Step six—Update the fitness values of the new positions;

Step seven—Repeat steps two to six until a stopping criterion is met.

## 4. Results and Discussion

### *4.1. System Parameters*

Figure 2 illustrates a multi-energy system that utilizes various devices to perform different energy conversion processes; the system has been implemented using the MATLAB software. Table 1 provides a comprehensive overview of the efficiency factors associated with different equipment categories involved in power supply, energy conversion, and energy storage processes. The efficiency factors specified in the table quantitatively represent the conversion efficiency achieved by each equipment type in their respective processes.

**Table 1.** Efficiency of each equipment.

| Part | Equipment | Efficiency Factor |
|------|-----------|-------------------|
| Supply | P2G | 0.6 |
| | Gas tank | 0.95 |
| Conversion | Bromine cooler | 1.38 |
| | Electric refrigerator | 3 |
| | Electric boiler | 3 |
| | Gas boiler | 7.92 |
| | Micro gas engine | 6.65 |
| Storage | Battery | 0.95 |
| | Thermal storage tank | 0.95 |

Figure 3 depicts the daily load curve for a typical day, showing the electrical, thermal, and cooling load data based on Reference [27], which act as a source of information on the energy demand patterns for a particular system or area. Figure 4 depicts the daily output of energy generated by wind turbines and solar panels.

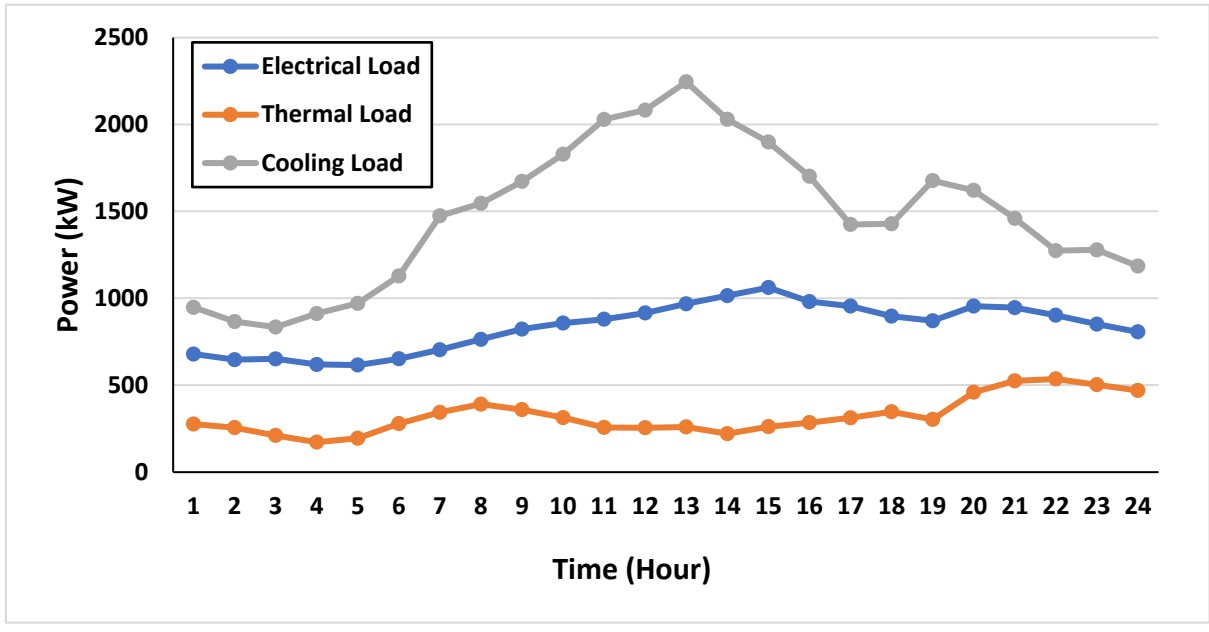

**Figure 3.** Typical daily load data in summer.

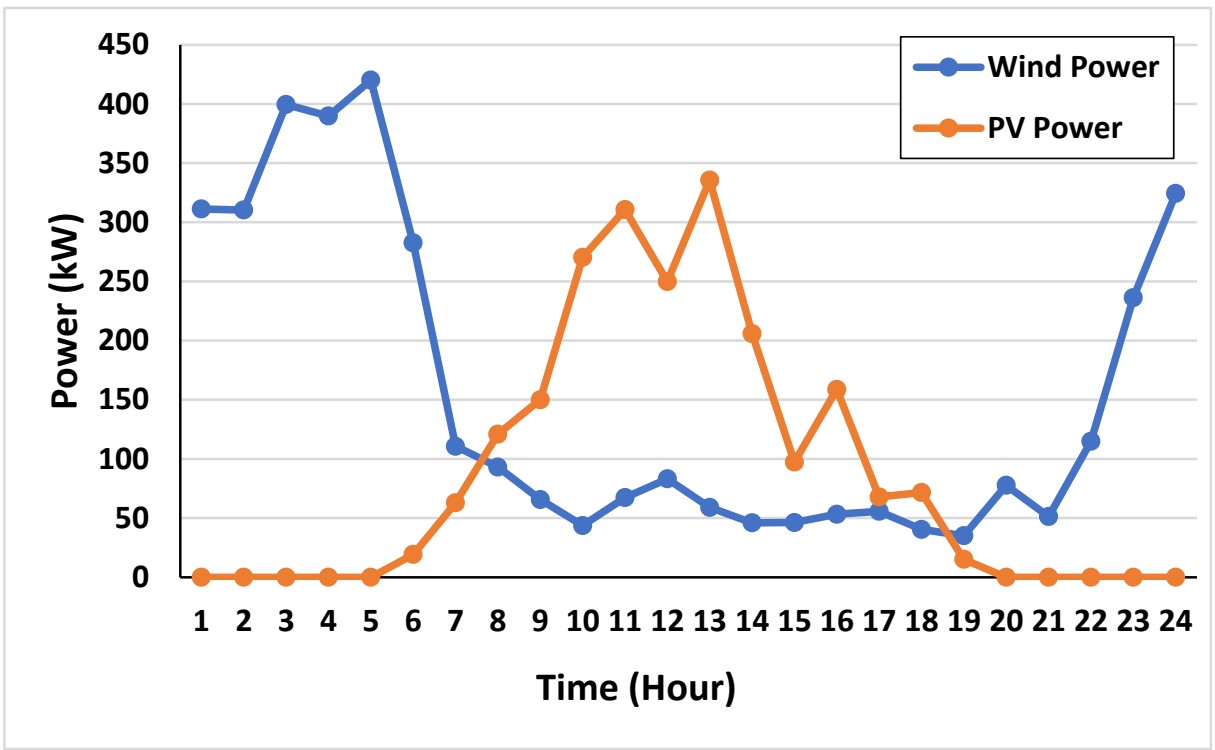

**Figure 4.** Maximum output of new energy.

Figure 5 shows the pricing structures for both electricity and natural gas. The daily electricity price includes both the purchase and sale price of electricity, with prices varying between peak and valley periods. In this figure, it is observed that the purchase price of electricity remains fixed throughout the day, indicating a constant cost for acquiring electrical energy. However, the selling price of electricity varies over time, reflecting fluctuations in the market and the varying demand for electricity during different periods of the day.

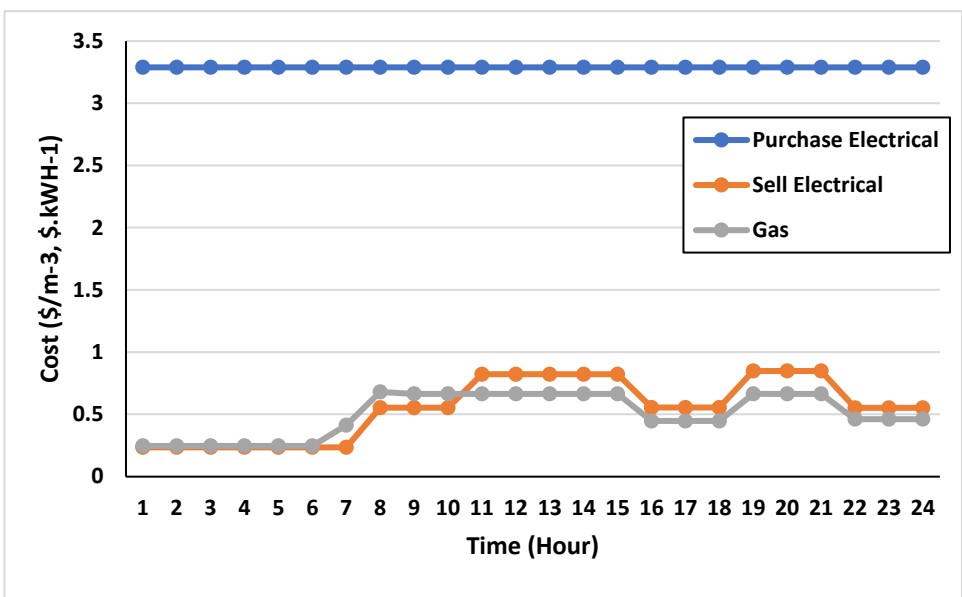

**Figure 5.** Energy prices.

*4.2. Analysis of Scheduling*

In the scheduling simulation analysis with a scheduling period of one day, the balance of supply and demand of four types of energy, namely, electricity, heat, cooling and gas, and the changes of energy output over time, were obtained. The simulation results show that the all-day wind turbine output and the photovoltaic output can be used to increase renewable energy output. The actual output of new energy is more affected by the peak and valley prices of electricity, which in turn affect the changes in output of the power grid and gas network, as shown in Figure 6. Figure 6 demonstrates the amount of gas required to meet the remaining power demand throughout the day. It provides valuable information regarding the contribution of natural gas in fulfilling the energy needs that are not met by wind and solar power sources.

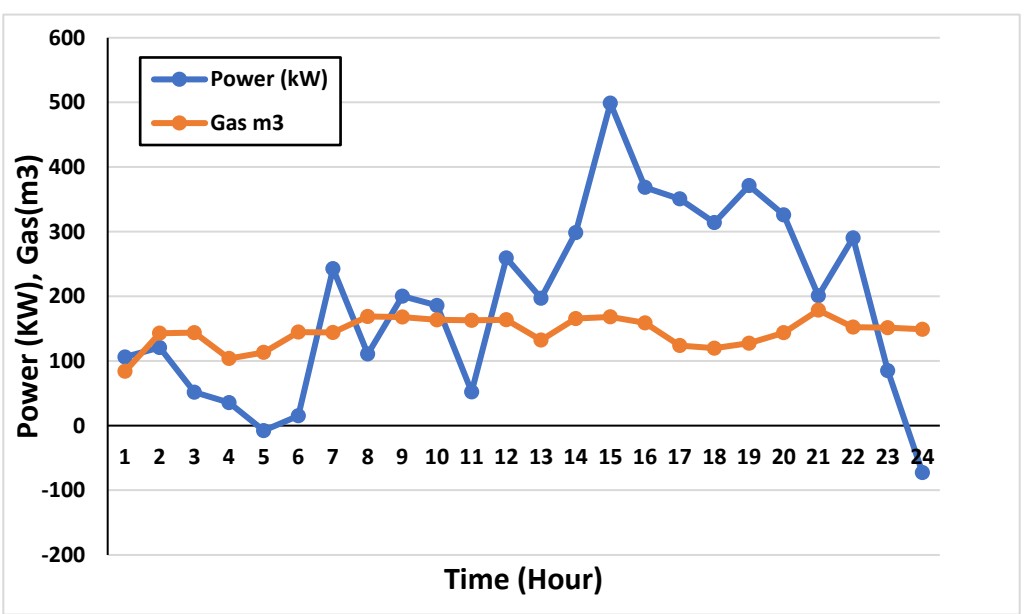

**Figure 6.** The output of the power grid and the natural gas grid.

Figure 6 shows how much gas is needed to meet the rest of the power demand during the day.

Figure 7a illustrates the balance of supply and demand in the four energy sources. In particular, during the valley price period (01:00–06:00), the electrical load is mainly contributed by the grid, and micro-fuel wind turbines and photovoltaic power can be used to satisfy the output, with a small amount of wind turbine output required to supplement it. During this period, the battery is charged at the flat electricity price (09:00–19:00, and 23:00–16:00 and 20:00–22:00), which relieves the peak load of electricity consumption and ensures the economy of the coordinated dispatch. Overall, the simulation results demonstrate the effectiveness of the collaborative scheduling approach in achieving a balance in the supply and demand of multiple energy sources while optimizing the economic efficiency of the energy system. At peak and flat electricity prices (09:00–24:00), the increase in cooling and heating loads leads to the activation of electrical equipment. Figure 7b shows that the micro-combustion engine is the primary source of thermal load, and a significant amount of heat is used for the refrigeration of the lithium bromide refrigerator. During the period of lowest electricity prices, the demand for heat load is low, and is primarily met by micro-gas turbines. In contrast, during peak and flat electricity price times, the demand for heat load increases, and micro-gas turbines work at full capacity, supplemented by gas boilers, electric boilers, and gas storage tanks. The coupling of equipment based on the difference in energy prices is realized here, reflecting its economic value. Furthermore, for cooling load, the lithium bromide refrigerator is the primary source, and electric refrigerators are responsible for supplementary amounts, as shown in Figure 7c. Lastly,

the supply of natural gas is mainly derived from the natural gas network, and P2G and gas storage tanks play a specific regulatory role, as shown in Figure 7d. The balance in the supply and demand of the four types of energy sources, i.e., electricity, heat, cooling, and gas, is ensured through collaborative scheduling simulation analysis, and changes in energy output over time are observed. The simulation results help us to achieve the goal of increasing renewable energy output by using the changes in the output of the power grid and gas network, as affected by the peak and valley prices of electricity, and the actual output of new energy.

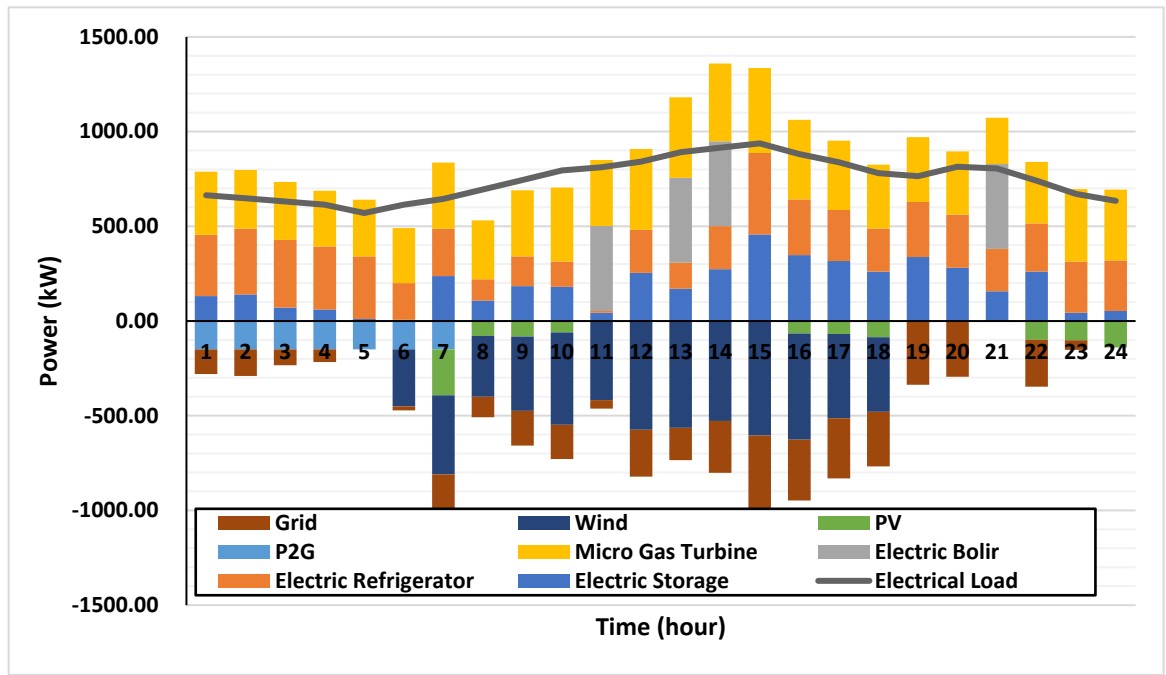

(**a**)

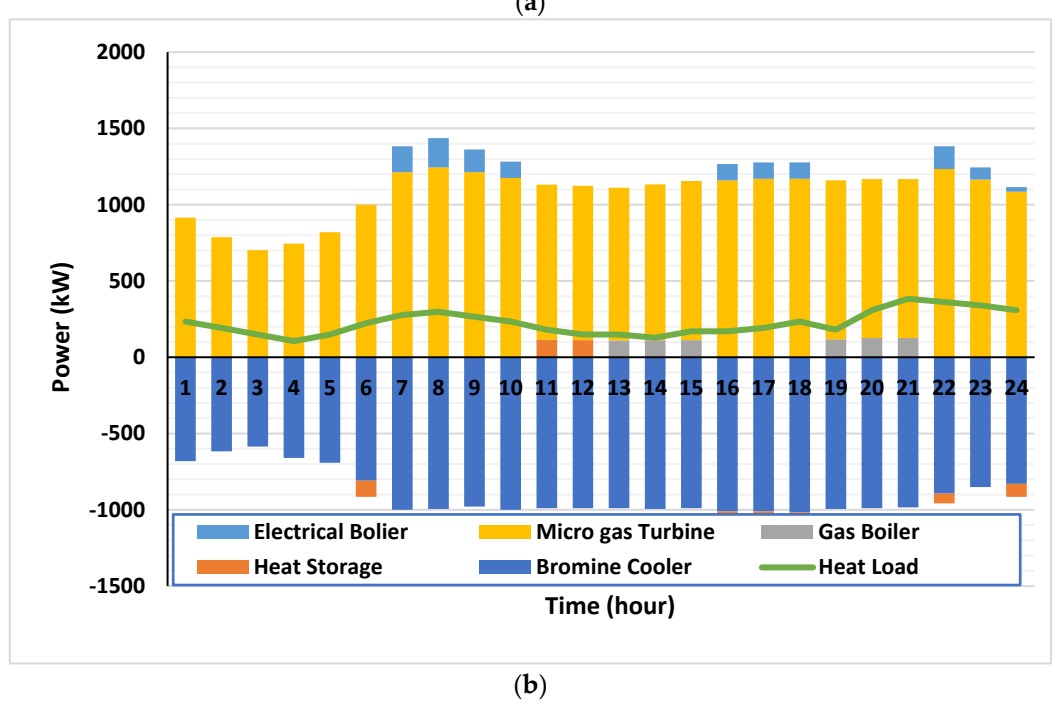

(**b**)

**Figure 7.** *Cont.*

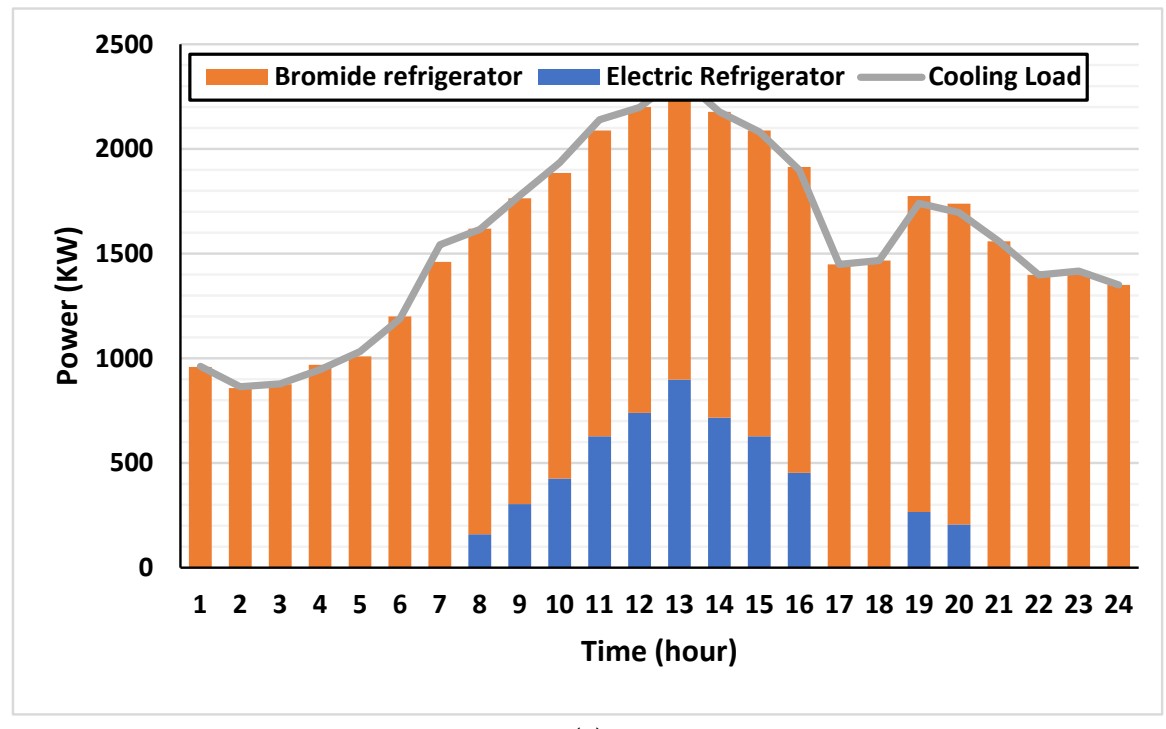

(**c**)

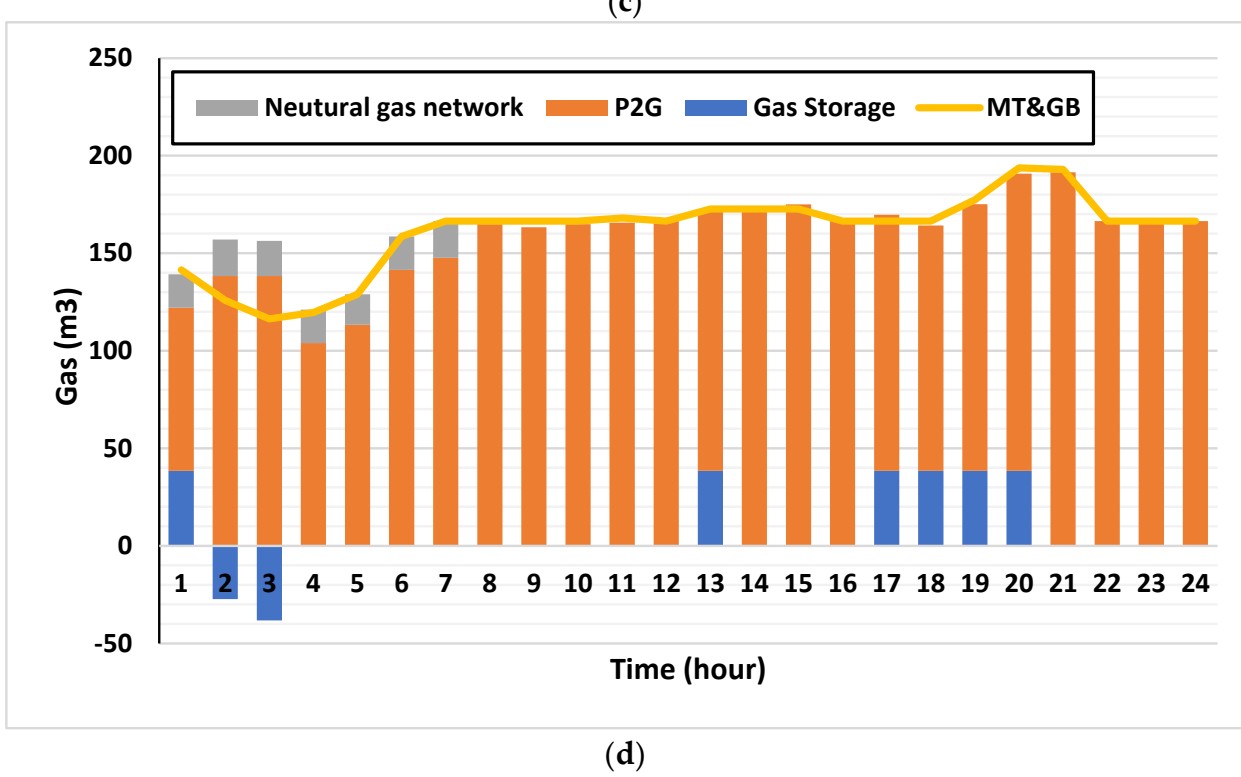

(**d**)

**Figure 7.** Balance of energy supply and demand. (**a**) Electricity; (**b**) thermal energy; (**c**) cooling energy; (**d**) natural gas.

### 4.3. Analysis of Collaborative Scheduling in Different Cases

The coordination scheduling of a hybrid energy system has been explored in three cases, wherein each case represents a different level of complexity of the energy hub. The scheduling period for each case is one day, and the associated scheduling costs are reported in Table 2.

**Table 2.** Comparison of associated scheduling costs.

| Module | Case 1 | Case 2 | Case 3 |
|---|---|---|---|
| Consumption module | √ | √ | √ |
| Conversion module | √ | √ | √ |
| Supply module | × | √ | √ |
| Storage module | × | × | √ |
| Dispatch cost/USD | 2774.91 | 2345.41 | 2238.47 |

Case 1 involves an energy hub that consists of conversion equipment only, which converts different energy sources into usable forms. The focus of this case is to test the coordinated scheduling of the energy hub.

In Case 2, a supply module is added to the energy hub, which enables it to absorb new energy. The main goal of this case is to examine the ability of the energy hub to coordinate the scheduling of the system and absorb new energy.

Case 3 presents the system constructed in this paper, where the energy hub includes four modules: supply, conversion, storage, and consumption. These modules are highly coupled, and enable the energy hub to manage energy more efficiently. This scenario aims to investigate the energy hub's ability to absorb new energy, manage energy economically, and coordinate the dispatch of the system.

Table 2 compares the scheduling costs for the three cases analyzed in the article, which comprise different choices of modules in the energy hub. Case 1 includes only equipment for conversion between energy sources, while Case 2 adds a supply module to test the system's ability to absorb new energy and its coordinated scheduling. Case 3 is the system constructed in the article, with highly coupled supply, conversion, storage, and consumption modules.

The table shows that Case 3 is the most economical, with savings of approximately 4.66% compared to Case 2 and 19.33% compared to Case 1. These results indicate the economic benefits of Case 3, which was enhanced with a supply module, facilitating the integration of renewable energy sources like wind or solar power that offer cost-effective and environmentally friendly energy options compared to conventional sources. Additionally, the inclusion of a storage module in Case 3 enables efficient energy management by storing excess energy during periods of low demand or high renewable energy generation.

We have analyzed the outputs of power grids and natural gas networks in the three cases, as shown in Figures 8 and 9. Figure 8 shows that the increases in supply modules in Case 2 and Case 3, along with the optimal cost of coordinated scheduling, lead to a decrease in grid output compared to Case 1. The increase in electrical conversion and P2G new energy consumption during low electricity prices contributes to this decrease. The difference between Case 2 and Case 3 is mainly reflected in the storage module. Figure 9 demonstrates that the storage module stores energy during flat electricity price periods and releases it during peak electricity price periods, which reduces electricity consumption during peak periods and plays a role in peak shaving and valley filling.

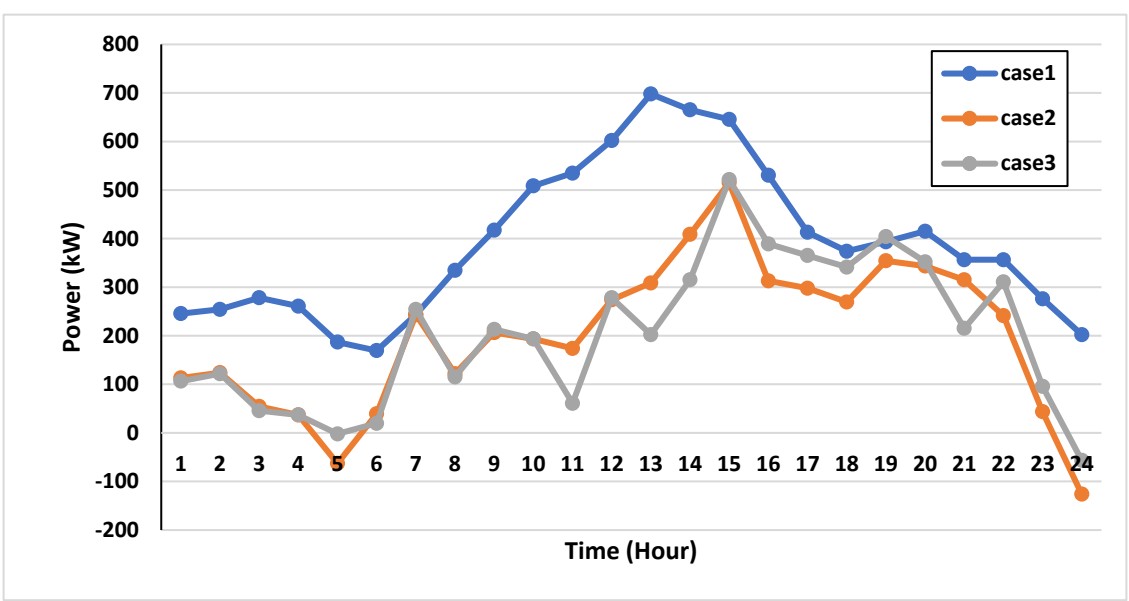

**Figure 8.** Grid output in different cases.

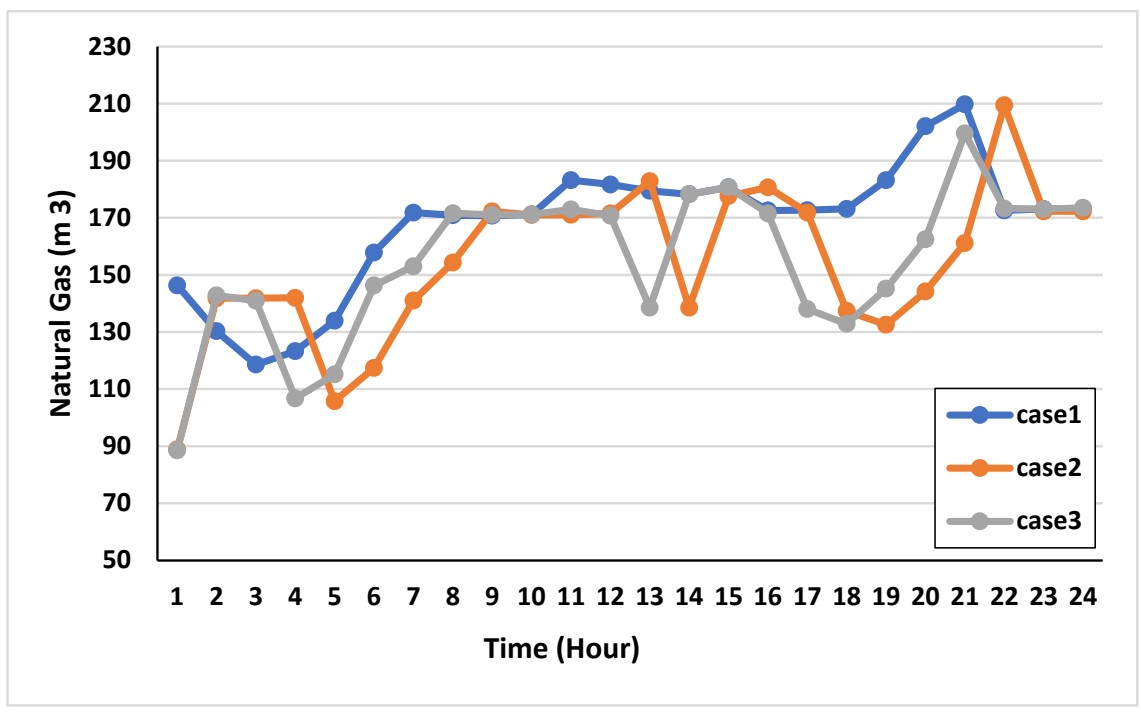

**Figure 9.** Natural gas network output in different cases.

Figure 9 shows that the main difference between Case 1, Case 2, and Case 3 arises with the inclusion of P2G equipment and a gas storage tank in the supply module. Additionally, the difference between Case 2 and Case 3 is small, and is mainly reflected in the utilization of a gas storage battery and a heat storage tank in the storage module. Therefore, the multi-energy system in Case 3 can significantly reduce the cost of coordinated scheduling, and lower the energy consumption during peak periods, providing peak shaving and valley filling benefits. Furthermore, it also contributes to the consumption of new energy and the electrical conversion of P2G equipment, highlighting the environmental and economic benefits of the P2G multi-energy system.

Figure 10 illustrates the final comparative results of Equation (14), including operating costs, emissions, and processing time using various algorithms. The parameters for both the Attractive Repulsive Shuffled Frog-Leaping (ARSFL) and Particle Swarm Optimization (PSO) algorithms are set as population size—50 (frogs or particles) and maximum number of iterations—100. The analysis specifically focuses on the ARSFL algorithm across different cases. In Case 3, the ARSFL algorithm successfully reduces the uncertainty of the energy rate by 5.77% compared to Case 2 and by 12.8% compared to Case 1. This improvement highlights the enhanced robustness of the model when incorporating scenic output and comprehensive demand response uncertainty, even at the expense of certain economic benefits. Furthermore, the ARSFL algorithm significantly optimizes the uncertainty of the energy purchase model in Case 3 compared to other algorithms. Specifically, it reduces the uncertainty by 0.64% and 1.3% when compared to the Particle Swarm Optimization (PSO) and Genetic Algorithm (GA), respectively. This demonstrates the superior performance of the ARSFL algorithm in managing and minimizing energy purchase uncertainty. Additionally, the execution time of the ARSFL algorithm is remarkably reduced compared to both PSO and GA. It saves 1.886 s (equivalent to a 1.59% reduction) compared to PSO and 3.117 s (equivalent to a 2.7% reduction) compared to GA. This efficiency of processing time further enhances the overall effectiveness and practicality of the ARSFL algorithm incorporating an attractive–repulsive mechanism that efficiently explores the search space and guides the optimization process towards promising solutions. This mechanism allows the algorithm to quickly converge towards optimal or near-optimal solutions, reducing the number of iterations required for convergence. The algorithm leads to a significant increase in the profit cost and total cost of the MES. Nevertheless, these costs are justified by the flexibility of the system in adapting to fluctuating real-time market electricity prices and operating costs.

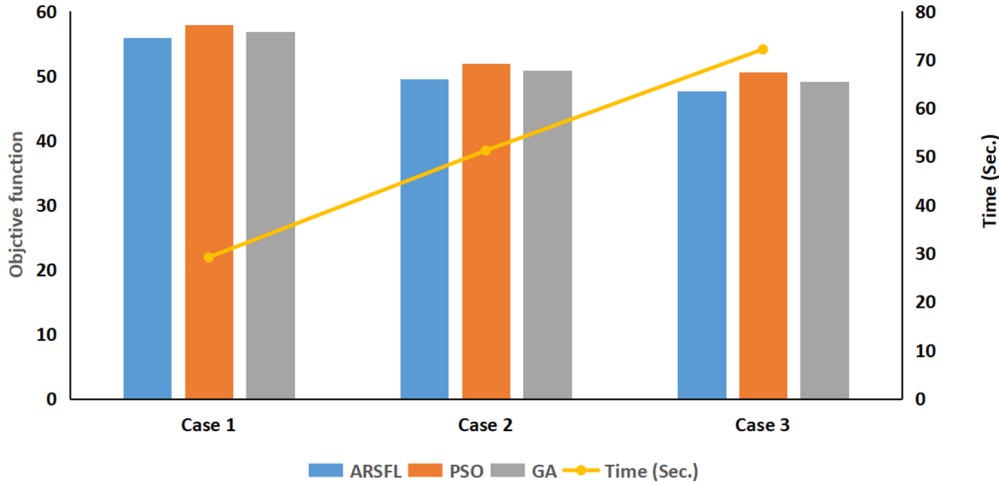

**Figure 10.** System optimization—comparison with other algorithms over 40 iterations.

## 5. Conclusions

This paper has proposed using the Attractive Repulsive Shuffled Frog-Leaping (AR-SFL) algorithm to optimize the scheduling of energy resources in the MES. The AR-SFL algorithm efficiently balances the conflicting objectives of minimizing the cost of energy production and reducing greenhouse gas emissions. The model's effectiveness was evaluated using real-world data from a university campus, and the results demonstrate that the proposed model can effectively reduce greenhouse gas emissions and energy costs while meeting energy demand requirements. The algorithm incorporates an attractive–repulsive mechanism that assigns attractive and repulsive forces to the frog population based on their fitness values. Future research could further investigate the scalability and robustness of the proposed model in different contexts, and explore additional strategies for promoting collaboration and inclusivity in energy management. Sensitivity analysis and validation

against a wider range of scenarios and data sets would help us determine the model's reliability, and identify potential weaknesses or limitations.

**Author Contributions:** Conceptualization, K.K. and O.I.A.; methodology, H.T. and M.N.M.; software, M.N.M.; validation, H.T. and M.N.M.; formal analysis, A.A.H.K.B.; investigation, K.K. and O.I.A.; writing—original draft preparation, O.I.A.; writing—review and editing, A.A.H.K.B. All authors have read and agreed to the published version of the manuscript.

**Funding:** Universiti Malaysia Pahang grant number [RDU223018] has played a crucial role in facilitating the execution of this research and the dissemination of its findings.

**Institutional Review Board Statement:** Not applicable.

**Informed Consent Statement:** Not applicable.

**Data Availability Statement:** Data is unavailable due to privacy.

**Acknowledgments:** We would like to express our gratitude to the individuals and organizations who have contributed to the successful completion of this research. Furthermore, we would like to acknowledge Universiti Malaysia Pahang and Gulf university, Bahrain.

**Conflicts of Interest:** The authors declare no conflict of interest.

## Nomenclature

| Term/Variable | Description |
| --- | --- |
| MES | Multi-energy systems |
| ARSFL | Attractive Repulsive Shuffled Frog-Leaping |
| P2G | Power-to-gas |
| PSO | Particle Swarm Optimization |
| GA | Genetic Algorithm |
| MILP | Mixed-Integer Linear Programming |
| $P_e^t$, $P_g^t$ | Output of electric energy and natural gas in the module |
| $P_e^{new}$, $P_g^{new}$ | Input of electric energy and natural gas in the module |
| $P_e^{new}$, $P_g^s$ | Input of new energy and gas storage tank |
| $W_g(t-1)$, $W_g(t)$ | Stored energy of the equipment before and after gas storage or deflation |
| $P_g^{ch}$, $P_g^{dis}$ | Energy stored in or released by the gas storage tank |
| $\eta_g^{ch}$, $\eta_g^{dis}$ | Gas storage and deflation efficiency |
| $\mu$ | Variable, with 1 representing the inflated state and 0 representing the deflated state |
| C | Coupling matrix |
| $P_e^T$, $P_c^T$, $P_h^T$ | Converters for electrical output, cold output, and heat output of the module |
| $\beta_1$, $\beta_2$, $\beta_3$ | Allocation coefficients for electric load, electric refrigerator, and electric boiler, with the sum equal to 1 |
| $\gamma$ | Proportion coefficient for a micro-gas turbine's natural gas consumption |
| $\delta$ | Proportion coefficient for total heat consumed by the refrigerator |
| $\eta_{AC}^c$ | Refrigeration coefficient for electric refrigerator |
| $\eta_{AR}^c$ | Refrigeration coefficient for lithium bromide refrigerator |
| $\eta_h^{EB}$ | Heating coefficient for electric boiler |
| $\eta_h^{GB}$ | Heating coefficient for steam boiler |
| $\eta_e^{MT}$, $\eta_h^{MT}$ | Micro-gas turbine's electrical efficiency and heating coefficient |
| $SOC(t)$ | State of charge of the battery at time t |
| $\vartheta$ | Discharge rate of the battery itself |
| $E_c$ | Rated capacity of the battery |
| $P_e^c$, $P_e^{dis}$ | Battery energy stored or released |
| $\eta_e^c$, $\eta_e^{dis}$ | Efficiency of charge and discharge |
| ci | Cost of purchasing electricity from the grid at time t |
| pit | Power purchased from the grid at time t |
| di | Cost of natural gas at time t |

| | |
|---|---|
| pht | Natural gas consumed for heating at time t |
| ei | Cost of hydrogen produced from P2G at time t |
| pgta | Hydrogen consumed for power generation at time t |
| fi | Cost of hydrogen stored in tanks at time t |
| pgtr | Hydrogen consumed for transportation at time t |
| gi | Cost of electricity exported to the grid at time t |
| pe | Penalty cost for not meeting the energy demand at time t |
| D(t) | Power demand at time t |
| ci | Cost of purchasing electricity from the grid at time t |
| Pit | Power purchased from the grid at time t |

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
