# Peer review of "Sustainable Green Energy Management: Optimizing Scheduling of Multi-Energy Systems Considered Energy Cost and Emission Using Attractive Repulsive Shuffled Frog-Leaping"

_sustainability, doi:10.3390/su151410775_

Round 1
Reviewer 1 Report
1. The quality of the abstract could be enhanced by providing some numerical results.
2. Many researchers proposed various optimization techniques for MES scheduling such as Mixed Integer Linear Programming (MILP), Particle Swarm Optimization (PSO), Fuzzy logic algorithm, Grey wolf algorithm, and Genetic Algorithms (GA). - Provide a few references for each algorithm. 3. Please avoid the sentences like: In [4] proposes a mathematical model for optimizing the operation of a multi-energy system consisting of electricity, gas, and heat networks. This can be rewritten as: A mathematical model for optimizing the operation of a multi-energy system consisting of electricity, gas, and heat networks was proposed by Li et al. [4]. Completely revise the literature review section. 4. Illustrate the research gap and novelty of your work. 5. Please write the equation numbers correctly. Equation number (i) is not seen. 6. Include a separate nomenclature section. 7. Explain the equations clearly. Some of the equations are difficult to understand. 8. Demonstrate the constraints used in section 3. 9. Section 3.2 is missing. 10. Include a separate section to illustrate the shuffled frog leaping algorithm. 11. What are the parameters of the proposed algorithm? This is important as the performance of any bio-inspired algorithm mainly depends on the parameters. 12. Please rewrite Table 1. No need to include the symbols in the table as you have already mentioned the efficiency factor. 13.Language editing is needed to improve the quality of the paper. 14. Write more about Table 1. 15. " The efficiency and coefficients provided in Table 1 are based on scientific principles and measurements, and represent the performance characteristics of each device under specific operating conditions" - I can see only the efficiency values not the coefficients. Please verify. 16. What is the relation between Figures 3 and 4? You have taken Figure 4 from a literature paper. What do you want to convey from Figure 4? Provide more details. 17. Include more explanation for Figures 5 and 6. 18. Check the caption of Table 2. You have mentioned Scheduling costs in the text (line number 419). But, in Table 1 it is presented as Dispatch cost/$. Which one is correct? Please check. 19. The table shows that Case 3 is the most economical, with a savings of approximately 425 4.66% compared to Case 2 and 19.33% compared to Case 1 - Why? Justify. 20. The performance of the proposed algorithm can be compared with other algorithms addressed in the literature. 21. What are the benefits of this work to the stakeholders? 22. What are the limitations of your work? Include in the conclusion section.Author Response
Dear Prof.,
We sincerely appreciate the valuable suggestions and the significant amount of time dedicated by the reviewer to provide their insightful feedback.
Please find below our revised response list:
- The quality of the abstract could be enhanced by providing some numerical results.
Answer: Kindly, check the modified abstract.
- Many researchers proposed various optimization techniques for MES scheduling such as Mixed Integer Linear Programming (MILP), Particle Swarm Optimization (PSO), Fuzzy logic algorithm, Grey wolf algorithm, and Genetic Algorithms (GA). - Provide a few references for each algorithm.
Answer: These reference been added.
[10] Handan Akulker, Erdal Aydin. Optimal design and operation of a multi-energy microgrid using mixed-integer nonlinear programming: Impact of carbon cap and trade system and taxing on equipment selections. Applied Energy, Volume 330, Part A, 2023, 120313.
[11] Ning Wang, Bo Li, Yan Duan, Shengling Jia. A multi-energy scheduling strategy for orderly charging and discharging of electric vehicles based on multi-objective particle swarm optimization. Sustainable Energy Technologies and Assessments, Volume 44, 2021, 101037.
[12] Wei Dong, Qiang Yang, Xinli Fang, Wei Ruan. Adaptive optimal fuzzy logic based energy management in multi-energy microgrid considering operational uncertainties. Applied Soft Computing. Volume 98, 2021, 106882.
[13] Chaoshun Li, Wenxiao Wang, Deshu Chen. Multi-objective complementary scheduling of hydro-thermal-RE power system via a multi-objective hybrid grey wolf optimizer. Energy, Volume 171, 2019, Pages 241-255.
[14] Sward J.A., T.R. Ault, K.M. Zhang. Genetic algorithm selection of the weather research and forecasting model physics to support wind and solar energy integration. Energy, Volume 254, Part B, 2022, 124367.
- Please avoid the sentences like: In [4] proposes a mathematical model for optimizing the operation of a multi-energy system consisting of electricity, gas, and heat networks. This can be rewritten as: A mathematical model for optimizing the operation of a multi-energy system consisting of electricity, gas, and heat networks was proposed by Li et al. [4]. Completely revise the literature review section.
Answer: Done, modified
- Illustrate the research gap and novelty of your work.
Answer: We add to introduction (There is a need for comprehensive and integrated optimization approaches for managing Multi-Energy Systems (MES) that consider the diverse nature of energy sources, uncertain and intermittent characteristics of renewable energy, and computational complexity). The proposed work introduces a collaborative scheduling model that utilizes the Attractive Repulsive Shuffled Frog-Leaping (ARSFL) algorithm. This model addresses the challenges associated with managing MES by providing an optimized approach to schedule energy resources efficiently.
- Please write the equation numbers correctly. Equation number (i) is not seen.
Answer: Done rechecked all equation numbers.
- Include a separate nomenclature section.
Answer: Done, Add in Table
Term/Variable |
Description |
MES |
Multi-Energy Systems |
ARSFL |
Attractive Repulsive Shuffled Frog-Leaping |
P2G |
Power-to-gas |
PSO |
Particle Swarm Optimization |
GA |
Genetic Algorithm |
MILP |
Mixed Integer Linear Programming |
, |
Output of electric energy and natural gas in the module |
, |
Input of electric energy and natural gas in the module |
, |
Input of new energy and gas storage tank |
Stored energy of the equipment before and after gas storage or deflation |
|
, |
Energy stored or released by the gas storage tank |
, |
Gas storage and deflation efficiency |
μ |
Variable, with 1 representing the inflated state and 0 representing the deflated state |
Coupling matrix |
|
,, |
Converters for electrical output, cold output, and heat output of the module |
,, |
Allocation coefficients for electric load, electric refrigerator, and electric boiler, with the sum equal to 1 |
|
Proportion coefficient for micro-gas turbine's natural gas consumption |
δ |
Proportion coefficient for total heat consumed by the refrigerator |
Refrigeration coefficient for electric refrigerator |
|
Refrigeration coefficient for lithium bromide refrigerator |
|
Heating coefficient for electric boiler |
|
Heating coefficient for steam boiler |
|
, |
Micro-gas turbine's electrical efficiency and heating coefficient |
State of charge of the battery at time t |
|
Discharge rate of the battery itself |
|
Rated capacity of the battery |
|
, |
Battery energy stored or released |
, |
Efficiency of charge and discharge |
ci |
Cost of purchasing electricity from the grid at time t |
pit |
Power purchased from the grid at time t |
di |
Cost of natural gas at time t |
pht |
Natural gas consumed for heating at time t |
ei |
Cost of hydrogen produced from P2G at time t |
pgta |
Hydrogen consumed for power generation at time t |
fi |
Cost of hydrogen stored in tanks at time t |
pgtr |
Hydrogen consumed for transportation at time t |
gi |
Cost of electricity exported to the grid at time t |
pe |
Penalty cost for not meeting the energy demand at time t |
D(t) |
Power demand at time t |
ci |
Cost of purchasing electricity from the grid at time t |
pit |
Power purchased from the grid at time t |
- Explain the equations clearly. Some of the equations are difficult to understand.
Answer: Done, recheck the details of introducing the equations.
- Demonstrate the constraints used in section 3.
Answer: Done, Constraints
- Section 3.2 is missing.
Answer: Sorry Prof., Is was format issue.
- Include a separate section to illustrate the shuffled frog leaping algorithm.
Answer: More details add to section 3.2
- What are the parameters of the proposed algorithm? This is important as the performance of any bio-inspired algorithm mainly depends on the parameters.
Answer: Add the The parameters for both the Attractive Repulsive Shuffled Frog-Leaping (ARSFL) and Particle Swarm Optimization (PSO) algorithms are set as Population Size: 50 (frogs or particles) and Maximum Number of Iterations: 100.
- Please rewrite Table 1. No need to include the symbols in the table as you have already mentioned the efficiency factor.
Answer: Modified
- Language editing is needed to improve the quality of the paper.
Answer: Done, Language check
- Write more about Table 1.
Answer: modified as (Table 1 provides a comprehensive overview of the efficiency factors associated with different equipment categories involved in power supply, energy conversion, and energy storage processes. The efficiency factors specified in the table quantitatively represent the conversion efficiency achieved by each equipment type in their respective processes. )
- " The efficiency and coefficients provided in Table 1 are based on scientific principles and measurements, and represent the performance characteristics of each device under specific operating conditions" - I can see only the efficiency values not the coefficients. Please verify.
Answer: Yes, the table show the efficiency values for each part. We modify the table title.
- What is the relation between Figures 3 and 4? You have taken Figure 4 from a literature paper. What do you want to convey from Figure 4? Provide more details.
Answer: By comparing Figures 3 and 4, one can analyze the correlation between energy demand and energy generation. Figure 4, show the typical amount of energy generated by wind turbines and solar panels at different times of the day.
- Include more explanation for Figures 5 and 6.
Answer: For figure 5 (In this figure, it is observed that the purchase price of electricity remains fixed throughout the day, indicating a constant cost for acquiring electrical energy. However, the selling price of electricity varies over time, reflecting fluctuations in the market and the varying demand for electricity during different periods of the day.) . For figure 6 (Figure 6 demonstrates the amount of gas required to meet the remaining power demand throughout the day. It provides valuable information regarding the contribution of natural gas in fulfilling the energy needs that are not covered by Wind and Solar power sources.). Actually, By analyzing Figure 6, stakeholders can assess the gas consumption patterns and understand the role of natural gas in supporting the overall power demand.
- Check the caption of Table 2. You have mentioned Scheduling costs in the text (line number 419). But, in Table 1 it is presented as Dispatch cost/$. Which one is correct? Please check.
Answer: Yes prof., Dispatch cost/$ is presented in Table 2.
- The table shows that Case 3 is the most economical, with a savings of approximately 425 4.66% compared to Case 2 and 19.33% compared to Case 1 - Why? Justify.
Answer: We modified as (The table shows that Case 3 is the most economical, with a savings of approximately 4.66% compared to Case 2 and 19.33% compared to Case 1. These results indicate that the economic benefits of Case 3 which enhanced with a supply module, facilitating the integration of renewable energy sources like wind or solar power which offers cost-effective and environmentally friendly energy options compared to conventional sources. Additionally, the inclusion of a storage module in Case 3 enables efficient energy management by storing excess energy during periods of low demand or high renewable energy generation.)
- The performance of the proposed algorithm can be compared with other algorithms addressed in the literature.
Answer: Yes, as shown in Figure 1.
- What are the benefits of this work to the stakeholders?
Answer:
The benefits of this work relies on the collaboration and acceptance of various stakeholders. By considering energy cost as one of the optimization objectives, the model helps stakeholders achieve cost-effective energy management.
Actually, By analyzing Figure 6, stakeholders can assess the gas consumption patterns and understand the role of natural gas in supporting the overall power demand
- What are the limitations of your work? Include in the conclusion section.
Answer: Done (Sensitivity analysis and validation against a wider range of scenarios and data sets would help determine the model's reliability and identify potential weaknesses or limitations.)

Reviewer 2 Report
This article contributes in a pertinent way to the science behind optimizing multiple-sources-based power systems with the SFL algorithm.
The language quality is clear and sound (very few typos that a quick spell-check could resolve). But I have identified a few items to further target the studies’ methodology and translate them into material clarification and potential recommendations for future research.
Line 64-101. Please consider putting this review of the different models that can be found in the literature in a specific section that touches on the current scientific and modeling landscape. It is not just an introduction; it is more like a mini-review. It is well done, and it deserves a specific space that you might want to put in section 2.
Line 102. Please add a definition of the Collaborative Scheduling Model since it is not yet a properly characterized or well-defined model, although its potential for being currently explored, in the digital era, by combining efforts among a set of entities to cooperate to solve some more complex scheduling problem. So, your definition of it is important and should be clarified in this paragraph.
Line 161-165. Please consider replacing “energy carriers” with “energy sources” and “energy usage” or explain which part of the associated energy carriers are considered for electricity (transport, distribution, etc.) or for natural gas (production, storage, etc.). Also, Heating and cooling are more associated with usage than carriers. This section needs to be clarified or re-explained in the context of your multiple inputs and requirements.
Line 189. Please identify the psch and pst terms that are used in equation 5.
Line 198. Please identify what differentiates the different b terms that are used in the equation.
Line 330. Please clarify what “rand4” is. Is it different than “rand” in equation 35 or is it just a typo?
Line 461-463 This is an important observation from your analysis: “Additionally, the execution time of the ARSFL algorithm is remarkably reduced compared to both PSO and GA. It saves 1.886 seconds (equivalent to a 1.59% reduction) compared to PSO and 3.117 seconds (equivalent to a 2.7% reduction) compared to GA.” More explanation on why you see that significant reduction in execution time would be appreciated from the readers' perspectives. Also, it deserves a short recall in the conclusions.
In the Conclusions section (Lines 483), you should outline the reduction in execution time. While you outline future research on that topic, you might give more insights on how the decision-makers could use the algorithm to reduce emissions and get better/sustainable energy management. The readers would welcome some practical orientations or recommendations from the authors.
As mentioned in the General Comments sections, the language quality is clear and sound (very few typos that a quick spell-check could resolve). But I have identified a few items to further target the studies’ methodology and translate them into material clarification and potential recommendations for future research.
Line 19. Please correct powertogas to one of, either Power-to-gas or Power to gas, which are the correct way to identify the concept.
Lines 48, 51, 55, etc.. Please correct “In [6]…” to “Reference [6]” or “[Ref. [6]”. Do the same with [7], [8], etc. in the following lines.
Line 289. “Constraint” in the list title should be plural “Constraints”.
Author Response
Dear Prof.,
We sincerely appreciate the valuable suggestions and the significant amount of time dedicated by the reviewer to provide their insightful feedback.
This article contributes in a pertinent way to the science behind optimizing multiple-sources-based power systems with the SFL algorithm.
The language quality is clear and sound (very few typos that a quick spell-check could resolve). But I have identified a few items to further target the studies’ methodology and translate them into material clarification and potential recommendations for future research.
Please find below our revised response list:
- Line 64-101. Please consider putting this review of the different models that can be found in the literature in a specific section that touches on the current scientific and modeling landscape. It is not just an introduction; it is more like a mini-review. It is well done, and it deserves a specific space that you might want to put in section 2.
Answer: Appreciate your support. The introduction modifed.
- Line 102. Please add a definition of the Collaborative Scheduling Model since it is not yet a properly characterized or well-defined model, although its potential for being currently explored, in the digital era, by combining efforts among a set of entities to cooperate to solve some more complex scheduling problem. So, your definition of it is important and should be clarified in this paragraph.
Answer: Introduction modified by adding (The model focuses on integrating various energy sources, such as renewable energy, power grids, and storage systems, while considering constraints and objectives related to energy cost, emission reduction, and sustainability. )
- Line 161-165. Please consider replacing “energy carriers” with “energy sources” and “energy usage” or explain which part of the associated energy carriers are considered for electricity (transport, distribution, etc.) or for natural gas (production, storage, etc.). Also, Heating and cooling are more associated with usage than carriers. This section needs to be clarified or re-explained in the context of your multiple inputs and requirements.
Answer: Done, modify replacing “energy carriers” with “energy sources”
- Line 189. Please identify the psch and pst terms that are used in equation 5.
Answer: We added a separate nomenclature section
- Line 198. Please identify what differentiates the different b terms that are used in the equation.
Answer: We added a separate nomenclature section
- Line 330. Please clarify what “rand4” is. Is it different than “rand” in equation 35 or is it just a typo?
Answer: Yes, modified as “rand()”
- Line 461-463 This is an important observation from your analysis: “Additionally, the execution time of the ARSFL algorithm is remarkably reduced compared to both PSO and GA. It saves 1.886 seconds (equivalent to a 1.59% reduction) compared to PSO and 3.117 seconds (equivalent to a 2.7% reduction) compared to GA.” More explanation on why you see that significant reduction in execution time would be appreciated from the readers' perspectives. Also, it deserves a short recall in the conclusions.
- Answer: We added (This efficiency in processing time further enhances the overall effectiveness and practicality of the ARSFL algorithm incorporates an attractive repulsive mechanism that efficiently explores the search space and guides the optimization process towards promising solutions. This mechanism allows the algorithm to quickly converge towards optimal or near-optimal solutions, reducing the number of iterations required for convergence. ).
- In the Conclusions section (Lines 483), you should outline the reduction in execution time. While you outline future research on that topic, you might give more insights on how the decision-makers could use the algorithm to reduce emissions and get better/sustainable energy management. The readers would welcome some practical orientations or recommendations from the authors.
Answer: Conclusion had been update.
Comments on the Quality of English Language
As mentioned in the General Comments sections, the language quality is clear and sound (very few typos that a quick spell-check could resolve). But I have identified a few items to further target the studies’ methodology and translate them into material clarification and potential recommendations for future research.
- Line 19. Please correct powertogas to one of, either Power-to-gas or Power to gas, which are the correct way to identify the concept.
Answer: Yes Prof. We modify powertogas to Power-to-gas
- Lines 48, 51, 55, etc.. Please correct “In [6]…” to “Reference [6]” or “[Ref. [6]”. Do the same with [7], [8], etc. in the following lines.
Answer: Yes Prof., We modify.

Reviewer 3 Report
The paper is written in a clear and concise manner, making it accessible to a broad readership.
The paper provides significant contributions to the field of sustainable green energy management by presenting a novel approach to optimizing the scheduling of multi-energy systems. It offers an important advancement in the current knowledge, especially in the use of ARSFL for energy scheduling. The paper has an overall merit for publication, but improvements can be made, particularly in discussing the methodology and design of the study in more depth.
Recommendations:
.Provide more details about how the attractive-repulsive mechanism works within the AR-SFL algorithm.
.The methodology and design could have been discussed more thoroughly. For example, the authors could have provided more details on how the ARSFL algorithm was implemented in the model, what specific constraints were considered, and how they were handled. A detailed explanation of the demand-response mechanism and how consumers can adjust their energy consumption patterns in response to price signals and other incentives would also have been beneficial.
.Discuss in more detail how the model can be scaled to larger or more complex systems.
.Share the raw data, if possible, to facilitate replication and further studies by other researchers.
. Review the numbering of the reference list that has duplicate numbering.
.Review the text that has errors that must be corrected, e.g. line 19 and 102.
The English language is appropriate and understandable, with minor room for improvement in some areas which could be corrected to further improve the paper's quality.
Author Response
Dear Prof.,
We sincerely appreciate the valuable suggestions and the significant amount of time dedicated by the reviewer to provide their insightful feedback.
The paper is written in a clear and concise manner, making it accessible to a broad readership.
The paper provides significant contributions to the field of sustainable green energy management by presenting a novel approach to optimizing the scheduling of multi-energy systems. It offers an important advancement in the current knowledge, especially in the use of ARSFL for energy scheduling. The paper has an overall merit for publication, but improvements can be made, particularly in discussing the methodology and design of the study in more depth.
Recommendations:
Please find below our revised response list:
- Provide more details about how the attractive-repulsive mechanism works within the AR-SFL algorithm.
Answer: more details add to section 3.2 (The AR-SFL algorithm utilizes an attractive-repulsive mechanism to guide the frogs towards better solutions and maintain population diversity. The attractive force attracts frogs to promising regions, promoting exploration and exploitation. Meanwhile, the repulsive force pushes frogs away from crowded or suboptimal regions, encouraging global exploration. By balancing these forces, the algorithm achieves an effective trade-off between exploration and exploitation, optimizing convergence speed and solution quality. This mechanism ensures efficient navigation of the search space, avoids premature convergence, and fosters population diversity.)
- The methodology and design could have been discussed more thoroughly. For example, the authors could have provided more details on how the ARSFL algorithm was implemented in the model, what specific constraints were considered, and how they were handled. A detailed explanation of the demand-response mechanism and how consumers can adjust their energy consumption patterns in response to price signals and other incentives would also have been beneficial.
Answer: The benefits of this work relies on the collaboration and acceptance of various stakeholders. By considering energy cost as one of the optimization objectives, the model helps stakeholders achieve cost-effective energy management. Also, by analyzing Figure 6, stakeholders can assess the gas consumption patterns and understand the role of natural gas in supporting the overall power demand
- Discuss in more detail how the model can be scaled to larger or more complex systems.
Answer: Scaling the model to larger systems requires seamless integration with existing energy management infrastructure, data sources, and decision-making processes.
- Share the raw data, if possible, to facilitate replication and further studies by other researchers.
Answer: We really appreciate any research collaboration in this field.
- Review the numbering of the reference list that has duplicate numbering.
Answer: All reference numbers check and modified.
- Review the text that has errors that must be corrected, e.g. line 19 and 102.
Answer: The text had been check and modified

Round 2
Reviewer 1 Report
Congratulations. The quality of the paper is improved now.